# Intersectional Fairness in Reinforcement Learning with Large State and Constraint Spaces

Eric Eaton [1] [*]   Marcel Hussing [1] [*]   Michael Kearns [1] [*]   Aaron Roth [1] [*]   Sikata Bela Sengupta [1] [*]   Jessica Sorrell [2] [*]

## Abstract

In traditional reinforcement learning (RL), the learner aims to solve a single objective optimization problem: find the policy that maximizes expected reward. However, in many real-world settings, it is important to optimize over multiple objectives simultaneously. For example, when we are interested in fairness, states might have feature annotations corresponding to multiple (intersecting) demographic groups to whom reward accrues, and our goal might be to maximize the reward of the group receiving the minimal reward. In this work, we consider a multi-objective optimization problem in which each objective is defined by a state-based reweighting of a single scalar reward function. This generalizes the problem of maximizing the reward of the minimum reward group. We provide oracle-efficient algorithms to solve these multi-objective RL problems even when the number of objectives is very large — for tabular MDPs, as well as for large MDPs when the group functions have additional structure. The contribution of this paper is that we are able to solve this class of multi-objective RL problems with a possibly *exponentially* large class of constraints over intersecting groups in both tabular and large state space MDPs in an oracle-efficient manner. Finally, we experimentally validate our theoretical results and demonstrate applications on a preferential attachment graph MDP.

## 1. Introduction

There are a number of reinforcement learning (RL) settings in which states correspond to either *individuals* or *groups*, each with their own properties, and actions correspond to the provision of goods or services among the states (Wen et al., 2021; Satija et al., 2023). For example, we could consider a road network of neighborhoods and the sequential distribution of disaster relief over the region (Li et al., 2024), a set of critical locations and touring these locations to provide surveillance, or hospital resources (e.g., beds, ventilators) and their allocation. In scenarios like these, what should we optimize for? If we optimize for a single objective, then the benefits of the actions may accrue disproportionately to some groups of people or locations over others. If there are natural groupings of states whose welfare we are concerned about (such as neighborhood demographics or disease sub-populations), then a natural fairness-motivated objective is *minimax* optimization: to maximize the reward that accrues to the group obtaining the minimum reward. Minimax objectives over groups have recently been studied in classification problems (Martinez et al., 2020; Diana et al., 2021), and have the attractive property that they *Pareto-dominate* solutions that seek to *equalize* reward across groups; every group will have higher reward in a minimax solution than they would in any equal reward solution. Problems like this are examples of *RL problems with constraints* (Calvo-Fullana et al., 2023; Dudík et al., 2020).

However, as noted by Kearns et al. (2018) (in a classification setting), especially in fairness motivated problems, it is insufficient to satisfy these kinds of constraints marginally. For example, suppose we have demographic groups defined by race, gender, and income, which are not mutually exclusive in that no attribute determines the value of any other. Satisfying constraints marginally on, e.g., racial groups does not guarantee that the corresponding constraints are satisfied on intersectional groups, defined by race and gender or gender and income. In general, if we have $d$ groupings of individuals, there are $2^d$ intersectional groups — an enormous number for even modestly large values of $d$.

In this paper, we study a constrained RL problem in the episodic setting that generalizes the problem of minimax group fairness, and is able to efficiently handle exponen-

[*] Equal contribution  [1] Department of Computer and Information Science, University of Pennsylvania, Philadelphia, USA [2] Department of Computer Science, Johns Hopkins University, Baltimore, Maryland, USA. Correspondence to: Sikata Sengupta <sikata@seas.upenn.edu>.

*Proceedings of the 42nd International Conference on Machine Learning*, Vancouver, Canada. PMLR 267, 2025. Copyright 2025 by the author(s).

tially many constraints — for example, those that arise from considering all intersections of $d$ grouping functions. We consider a problem in which the states $s \in \mathcal{S}$ of an MDP are annotated by feature vectors (e.g., describing the demographics of a population corresponding to state $s$). There is a collection of *group functions* $\mathcal{G}$ containing functions $g : \mathcal{S} \to \{0, 1\}$ indicating (as a function of the features at a state $s$) whether $s$ is a member of group $g$ or not. The groups can be arbitrarily intersecting. Every state/action pair $(s, a)$ is associated with a reward $r(s, a)$ that accrues to each group $g$ such that $g(s) = 1$. Given a policy $\pi$, the expected reward that accrues to group $g$ is $V^g(\pi) = \mathbb{E}_\pi[\sum_{t=1}^{H} r(s_t, a_t)g(s_t)]$, and the group-wise minimax reward maximization problem is to solve $\max_\pi \min_g V^g(\pi)$. Our goal is to give oracle-efficient algorithms for solving this problem over large state and constraint spaces — i.e., computational reductions to *unconstrained* RL problems and optimization problems over $\mathcal{G}$.

## 1.1. Our Results

We give three main results:

1. For MDPs with only polynomially many states — i.e., the *tabular* setting — we give an efficient reduction to a linear optimization oracle over $\mathcal{G}$ that optimally solves the constrained RL problem with constraints defined by $\mathcal{G}$, independently of the cardinality of $\mathcal{G}$. Here the oracle we need is for a batch/offline optimization problem, which is substantially easier than the RL/control problem. See Section 3.1.

2. For large MDPs, (which we cannot solve efficiently in the worst case, even in the standard single-objective setting), we give an efficient reduction from the problem of optimally solving the contrained RL problem with constraints from $\mathcal{G}$ to two problems: 1) the problem of finding the optimal policy in the standard (unconstrained) single-objective RL problem, and 2) the batch/offline optimization problem over $\mathcal{G}$. Here we need to assume that $\mathcal{G}$ has special structure — namely that it has a polynomial-sized *separator set* (Dudík et al., 2020) — but fortunately the class of boolean conjunctions that define the $2^d$ "intersectional" constraints given $d$ groupings of the data have this structure. See Section 3.2.

3. For arbitrary grouping functions $\mathcal{G}$ (which may not have small separator sets), we give an algorithm that makes iterative calls to a standard RL algorithm for the unconstrained problem and an optimization oracle over $\mathcal{G}$ that provably converges to the optimal constrained policy — albeit without a polynomial time convergence bound. See Section 3.3. Despite the lack of a polynomial-time convergence bound (in contrast to the

algorithm we give in Section 3.2), this algorithm has the advantage of simplicity (and not requiring separator set structure on $\mathcal{G}$), and is what we use in experiments; we find that empirically it works quite well.

In the above results, what we actually find is a *distribution* over policies that are minimax-optimal in expectation. But, this naively means that for any fixed policy in the support of our distribution, the constraints might be badly violated. This would be concerning in settings in which we care about a single episode rather than the average over many episodes and has a simple fix, described in **Appendix A.4**. Informally, we replace constraints of the form $g(x) \leq 0$ with constraints of the form $\max\{0, g(x)\} \leq 0$ which preserves convexity and eliminates the possibility that "slack" in the constraints on some policies cancel out in expectation with constraint violation in others. This is because by taking only the positive part of the constraint violation, we penalize our solution for constraint violation without rewarding it for slack.

We evaluate FairFictRL (Algorithm 4) on Barabási-Albert graphs (Barabási & Pósfai, 2016) with groups assigned based on the degree distribution of nodes. We show that our algorithm converges efficiently to a solution with low average constraint violations for all groups, while still optimizing the global objective. See Section 4.

## 1.2. Related Work

### 1.2.1. FAIR REINFORCEMENT LEARNING

There are many notions of fairness in RL that are distinct from what we study in this paper. Jabbari et al. (2017) define a notion of fairness that requires that the algorithm never play one action with higher probability over another unless the long-term reward of the optimal policy after playing the first action is higher than the second. Cousins et al. (2024) consider a welfare-centric notion of fair RL that encompasses a broad class of functions over a set of beneficiaries. They develop a model of adversarially-fair KWIK (knows-what-it-knows learning) and provide the algorithm $E^4$ (equitable $E^3$). It is important to note that their results mainly hold only for the tabular MDP setting. Michailidis et al. (2024) use a flexible form of Lorenz dominance to ensure a more equitable distribution of rewards and empirically demonstrate success of their method for real-world transport planning problems. Wen et al. (2021) consider model-based and model-free approaches to constrained fair RL problems and study these in the context of a bank offering loans to individuals. However, these methods do not extend for a very large number of overlapping groups. Satija et al. (2023) consider demographic fairness by constraining the group pair-wise difference in performances and provide theoretical guarantees for their algorithms in the tabular setting with an

empirical demonstration of success for larger state spaces. See Reuel & Ma (2024) for a survey of this literature. Aside from our differing objective, we differentiate from this literature by giving algorithms and theory that are able to handle both 1) a very large number of intersecting groups, and 2) provable guarantees beyond the tabular setting.

### 1.2.2. CONSTRAINED REINFORCEMENT LEARNING

Calvo-Fullana et al. (2023) consider the problem of constrained RL, especially for continuous state and action spaces. In their setting, they have $m$ reward functions and optimize global cumulative reward subject to each of the $m$ value functions for the given policy being at least as large as some threshold. To do this, they constructed augmented state-space MDPs and formulate the corresponding Lagrangian (with regularization) of this optimization problem. They provide an efficient Primal-Dual algorithm (using different scalarized reward optimization for the learner and online gradient descent (no-regret) for the regulator). Similar to Calvo-Fullana et al., Müller et al. (2024) provide Primal-Dual algorithms and construct their Lagrangian with regularization and optimistic exploration. They prove last-iterate convergence of their algorithm, enabling them to avoid error cancellations. Ding et al. (2023) also prove non-asymptotic convergence of their policy iterates for their policy-gradient-based primal-dual algorithms via regularization or optimism.

Miryoosefi et al. (2019) also study the problem of RL with convex constraints similarly observing a $d$-dimensional reward vector rather than a scalar. They also formulate this problem as a zero-sum game between a learner and regulator playing best-response vs. online gradient descent (no-regret). These algorithms require enumerating the constraints; in contrast we give "oracle efficient" algorithms that require only optimizing over the constraints, and hence can handle extremely large collections of constraints efficiently.

### 1.2.3. MULTI-OBJECTIVE CLASSIFICATION PROBLEMS

Agarwal et al. (2018) consider the problem of fairness in the binary classification setting and reduce this problem to a sequence (in a repeated zero-sum game between a learner and regulator) of cost-sensitive classification problems to provide a randomized classifier with low general error while (in expectation) satisfying the constraints. Kearns et al. (2018) extend Agarwal et al. (2018) to settings with a very large number of overlapping groups. They propose statistical notions of fairness that take into account "fairness gerrymandering" over this large number of subgroups and reduce their problem to a sequence of weak agnostic learning problems. We extend this style of algorithm from the classification to the RL setting.

## 2. Model and Preliminaries

We consider an episodic fixed-horizon Markov decision process (MDP) (Puterman, 2014) which can be formalized as a tuple $\mathcal{M} = (\mathcal{S}, \mathcal{A}, P_h, r_h, \mu)$, where $\mathcal{S}$ is the set of states, $\mathcal{A}$ is the set of actions, $H$ is the horizon, $r_h : \mathcal{S} \times \mathcal{A} \to [0, 1]$ is the reward function at time $h$, $P_h$ specifies the transition dynamics at time $h$, and $\mu$ is the initial state distribution. For simplicity, and without loss of generality, we assume rewards and transition dynamics are time-invariant, and denote them by $r$ and $P$ respectively. Without loss of generality, we assume that rewards are bounded within $[0, 1]$.

Throughout the paper, $[N]$ will denote the set $\{0, ..., N-1\}$. We will indicate a sequence of elements $s_1, s_2, \ldots, s_t$ by $s_{1:t}$. We write $x \sim \mathcal{U}(S)$ to denote sampling from the uniform distribution over set $S$, and $\nu \sim \text{Lap}(\rho)$ to denote sampling from the Laplace distribution with parameter $\rho$.

In the beginning, an initial state $s_0$ is sampled from $\mu$. At any time $h \in [H]$, the agent is in some state $s_h \in \mathcal{S}$ and chooses an action $a_h \in \mathcal{A}$ based on a function $\pi_h$ mapping from states to distributions over actions $\Pi : \mathcal{S} \mapsto \Delta\mathcal{A}$. As a consequence, the agent traverses to a new next state $s_{h+1}$ sampled from $P(\cdot|s_h, a_h)$ and obtains a reward $r(s_h, a_h)$. The sequence of functions $\pi_h$ used by the agent is referred to as its *policy*, and is denoted $\pi = \{\pi_h\}_{h \in [H]}$. A *trajectory* is the sequence of (state, action) pairs $\{(s_h, a_h)\}_{h \in [H]}$ taken by the agent over an episode of length $H$. In standard RL, the goal of the learner is to maximize the expected cumulative reward $\mathbb{E}_{s_0 \sim \mu, P}[\sum_{t=0}^{H-1} r(s_t, a_t)]$ over episodes of length $H$. We further define the value function as the expected cumulative return of following some policy $\pi$ from some state $s$ as $V^\pi(s) = \mathbb{E}_{s_0 \sim \mu_0, P}[\sum_{t=0}^{H-1} r(s_t, a_t)|\pi, s_0 = s]$. Due to the finite horizon of the episodic setting, we will also need to refer to the expected cumulative reward from state $s$ under policy $\pi$ from time $h \in [H]$. We denote this time-specific value function by $V_h^\pi(s) = \mathbb{E}_P[\sum_{t=h}^{H-1} r(s_t, a_t)|\pi, s_h = s]$. For the remainder of the paper, when indices for horizon $h$ are not listed, assume $h = 0$.

We consider a setting in which each state is assigned a feature vector $x \in \mathcal{X}$ over $d$ attributes. Alternatively, we can directly think of states as feature vectors themselves. We then define a set of functions $\mathcal{G}$, where each function $g \in \mathcal{G}, g : \mathcal{X} \to \{0, 1\}$ represents membership in the group $\{x \in \mathcal{X} : g(x) = 1\}$.

We denote the feature vector associated with state $s$ by $x(s)$, and use the shorthand $g(s_t) = g(x(s_t))$. Then, for each state visited, the scalar reward $r(s, a)$ will be given to all groups of which $x$ is a member. So, at time $t$, group $g$ receives $r(s_t, a_t)g(s_t)$. Note that, more generally, we could have a reward function defined over features and actions $r(x, a)$ and think about $r(x, a)g(x)$ as the corresponding

reward to the groups to which $x$ belongs.

Now, we can formulate our problem in terms of a repeated zero-sum game between a learner and a regulator. The learner is trying to find a policy from a class of policies $\Pi$ to optimize the expected cumulative reward (across all groups) and the regulator is trying to minimize the learner's payoff by finding groups with the lowest expected cumulative reward for the selected policy.

Let $V^g(\pi) = \mathbb{E}_{s\sim\mu,P,\pi}[\sum_{h=0}^{H-1} r(s_h,a_h)g(s_h)|s_0 = s]$ and note that it is equivalent to the average reward formulation $V^g(\pi) = H\,\mathbb{E}_{t\in[H]}\,\mathbb{E}_{P,\pi,\mu}[r(s_t,a_t)g(s_t)]$ and $V^{tot}(\pi) = H\,\mathbb{E}_{t\in[H]}\,\mathbb{E}_{P,\pi,\mu}[r(s_t,a_t)]$. We can consider the following *minimax reward* problem:

$$
\begin{aligned}
\max_{D\in\Delta\Pi}\quad & \mathbb{E}_{\pi\sim D}[V^{tot}(\pi)] \\
\text{subject to}\quad & \mathbb{E}_{\pi\sim D}[V^g(\pi)] \geq \alpha, \quad \forall g\in\mathcal{G} .
\end{aligned}
\tag{1}
$$

We call the formulation of Problem 1 a *minimax reward* problem because the solution that optimizes global cumulative reward subject to those constraints still is a member of the set of minimax solutions to the problem:

$$
\max_{D\in\Delta\Pi}\min_{G\in\Delta\mathcal{G}}\mathbb{E}_{\substack{\pi\sim D\\g\sim G}}[V^g(\pi)]
\tag{2}
$$

which can be solved by setting the global objective to be an optimization over $\alpha$, as mentioned below. It is worth noting that while the ellipsoid method ordinarily might have been able to handle an exponential number of constraints itself, given the large dimension of the primal player's action space it may be computationally inefficient for our class of problems. In order to solve this class of problems, we first formulate their Lagrangians. The resulting Lagrangian of Problem 2 is:

$$
\mathcal{L}(D,\lambda) = \mathbb{E}_{\pi\sim D}\left[V^{tot}(\pi) + \sum_{g\in\mathcal{G}}\lambda^g(V^g(\pi)-\alpha)\right] .
$$

We make a couple of observations. First, notice that for $\alpha = 0$ this problem always has a feasible solution. Moreover, if we make the objective of the optimization problem $\alpha$ rather than $V^{tot}$, then this becomes analogous to the minimax formulation in Problem 2. Additionally, we could always choose to set $\alpha_g$'s that are tailored to the individual groups rather than using a single global value $\alpha$ across all groups. Finally, we can use this framework to solve a more general class of multi-group constrained RL problems with constraints that are linear (and sometimes even convex) in the distribution over policies $D$. We can define the average reward objective of this problem as follows:

$$
U(D,\lambda) = \mathbb{E}_{\substack{\pi\sim D\\t\in[H]\\P,\pi,\mu}}[r(s_t,a_t) + \sum_{g\in\mathcal{G}}\lambda^g(r(s_t,a_t)g(s_t)-\tfrac{\alpha}{H})]
$$

Defining the following compact convex set $\Lambda$

$$
\Lambda = \left\{\lambda\in\mathbb{R}_+^{|\mathcal{G}|} : ||\lambda||_1 \leq C\right\} ,
$$

the resulting minimax problem under this formulation would be $\max_{D\in\Delta\Pi}\min_{\lambda\in\Lambda}U(D,\lambda)$ . Notice that in both cases, since the action spaces of both players are compact and convex, and the objective is affine in both variables, the conditions necessary for Sion's minimax theorem hold (Sion, 1958).

**Definition 2.1** (Regulator's Regret). For a given transcript of $\{D_t,\lambda_t,U(D_t,\lambda_t)\}_{t=1}^H$, we define the Adversary's regret to be:

$$
\sum_{t=1}^H\mathbb{E}\left[U(D_t,\lambda_t)\right] - \min_{\lambda\in\Lambda}\sum_{t=1}^H\mathbb{E}\left[U(D_t,\lambda)\right] .
$$

**Definition 2.2** ($\nu$-approximate minimax equilibrium). $(\hat{D},\hat{\lambda})$ is a $\nu-$approximate minimax equilibrium if :

$$
U(\hat{D},\hat{\lambda}) \leq \min_{\lambda\in\Lambda}U(\hat{D},\lambda) + \nu ,
$$

$$
U(\hat{D},\hat{\lambda}) \geq \max_{D\in\Delta\Pi}U(D,\hat{\lambda}) - \nu .
$$

**Definition 2.3** (Regulator's Best Response Function).

$$
Best_\lambda(D) = \begin{cases} 0 & \text{if nothing violated} \\ Ce_k \end{cases} ,
\tag{3}
$$

where $k$ is the index of the largest constraint violated. When the number of constraints is extremely large, this best response function may need to invoke an Optimization Oracle as defined in Subsection 2.1.

For each given $\lambda$ selected by the regulator, we can construct the transformed scalarized reward function for the learner as follows:

$$
r_\lambda(s_t,a_t) = \left(r(s_t,a_t) + \sum_{g\in\mathcal{G}}\lambda^g\left(r(s_t,a_t)g(s_t)-\tfrac{\alpha}{H}\right)\right) .
$$

To see this:

$$
\begin{aligned}
\mathbb{E}_{s_t,a_t\sim\pi}&\left[\sum_{t=1}^H r_\lambda(s_t,a_t)\right] \\
&= \mathbb{E}_{s_t,a_t\sim\pi}[\sum_{t=1}^H(r(s_t,a_t) + \sum_{g\in\mathcal{G}}\lambda^g(r(s_t,a_t)g(s_t)-\tfrac{\alpha}{H}))] \\
&= \mathbb{E}_{s_t,a_t\sim\pi}[\sum_{t=1}^H r(s_t,a_t) + \sum_{g\in\mathcal{G}}\lambda^g(\sum_{t=1}^H r(s_t,a_t)g(s_t)-\alpha)] \\
&= V^{tot}(\pi) + \sum_{g\in\mathcal{G}}\lambda^g(V^g(\pi)-\alpha) .
\end{aligned}
\tag{4}
$$

Thus, we can define the MDP $M_\lambda(S,A,P,\mu,r_\lambda)$ with the modified rewards and maximizing

$$
\mathbb{E}_{\pi\sim D}\left[\mathbb{E}_{s_t,a_t\sim\pi,P}\left[\sum_{t=1}^T r_\lambda(s_t,a_t)\right]\right]
$$

will provide a policy which we can represent the selection of with distribution variable $D$ that will also minimize the objective. Recall that in sequential play, the second player can achieve the value of the game via deterministic response. We once again can define the corresponding function purely in terms of feature vectors $x$ rather than purely in terms of states.

## 2.1. Oracles

The learner and regulator will make use of the oracles defined below. While we state them as strict optimizers, we observe that $\epsilon$-approximate versions of these oracles suffice for our applications.

**Definition 2.4** (Lin-OPT Oracle). A linear optimization oracle for $\mathcal{G}$, Lin-OPT$(c) = \arg\min_{g \in \mathcal{G}} \langle g_{\mathcal{S}}, c \rangle$, where $g_{\mathcal{S}}$ denotes the vector $(g(s_1), \ldots, g(s_{|\mathcal{S}|}))$.

**Definition 2.5** (OPT Oracle). An optimization oracle for $\mathcal{G}$, OPT$(s_{1:t}, c) = \text{argmin}_{g \in \mathcal{G}} \langle g_{s_{1:t}}, c \rangle$, where $g_{s_{1:t}}$ denotes the vector $(g(s_1), \ldots, g(s_t))$.

**Definition 2.6** (Learner's Best Response Oracle). We define $\mathcal{O}(U, \lambda)$ to be the best response oracle for the learner on input MDP $\mathcal{M}_\lambda$. That is, it returns $\arg\max_D U(D, \lambda)$, where $D$ is a point mass distribution, concentrated on a single policy $\pi$. We will frequently use the notation $\pi_t$ to refer to the policy on which a best response $D_t$ is supported. We use $\mathcal{O}_\epsilon(U, \lambda)$ to denote $\epsilon$-approximate best response.

The learner's best response oracle captures the assumption that for a given scalar reward, one is able to run a standard RL algorithm to learn an optimal policy for that given MDP. Note that in the tabular MDP setting this assumption can be realized by existing efficient algorithms for learning near-optimal policies, such as $E^3$ (Kearns & Singh, 2002) or R-max (Brafman & Tennenholtz, 2002) .

# 3. Algorithms and Theoretical Results

We will present our theoretical results in the following manner. For the settings of tabular MDPs and large state spaces with structured groups, we will first provide a general algorithm and analysis in terms of an arbitrary no-regret algorithm used by the regulator. We will then provide the specific regret analysis corresponding to each no-regret algorithm/setting used by the regulator. Finally, for the setting of large state spaces with general groups, we will provide a different algorithm and provide corresponding analysis. We use a modified version of the algorithm presented in (Agarwal et al., 2018) in Algorithm 1.

**Theorem 3.1.** *(Informal)[No-Regret Player Guarantee] For a sequence of best-response policies $(D_1, ..., D_T)$ and Lagrangian weights $(\lambda_1, ..., \lambda_T)$ maintained by the learner*

---

**Algorithm 1** MORL-BRNR (Multi-Objective RL–Approximate Min-Max RL Algorithm)

1: **Input:** bound $C$, best-response error $\epsilon$, $\mathcal{G}$ groups, $r$ reward function, access to MDP $\mathcal{M}$
2: Initialize $\lambda_0^g = 0$
3: **for** $t = 1, \ldots, T$ **do**
4: $\quad D_t \in \mathcal{O}_\epsilon(\cdot, \lambda_{i,t-1})$
5: $\quad \lambda_{t+1} = $ no-regret-update$(D_t, \lambda_t, C)$ *note that this will run one iteration of the subsequent no-regret algorithms provided*
6: $\quad \hat{D}_t = \frac{1}{t} \sum_{t'=1}^{t} D_{t'}; \hat{\lambda}_t = \frac{1}{t} \sum_{t'=0}^{t} \lambda_{t'}$
7: **end for**
8: **return** $(\hat{D}_T, \hat{\lambda}_T)$

---

*and regulator in MORL-BRNR [Algorithm 1],*

$$\sum_{t=1}^{T} \mathop{\mathbb{E}}_{\pi \sim D_t} [U(D_t, \lambda_t)] - \min_{\lambda \in \Lambda} \sum_{t=1}^{T} \mathop{\mathbb{E}}_{\pi \sim D_t} [U(D_t, \lambda)]$$
$$\leq reg(T, C, \gamma) \ ,$$

*where $reg(T, C, \gamma)$ is sublinear in $T$.*

In this paper, we will provide this form of a regret guarantee for the regulator with two different algorithms.

**Theorem 3.2** (Approximate Min-Max). *For a sequence of distributions over best-response policies $(D_1, ..., D_T)$ and Lagrangian weights from no-regret updates $(\lambda_1, ..., \lambda_T)$ maintained by the learner and regulator, MORL-BRNR [Algorithm 1] returns a $\nu$-approximate saddle point solution $(\hat{D}, \hat{\lambda})$ of $U$.*

*Proof.* Provided in Appendix A.1. □

## 3.1. Tabular MDPs

In the case of tabular MDPs, we adapt the algorithm Follow the Perturbed Leader (FTPL) given by (Kalai & Vempala, 2005) to our learning setting.

**Lemma 3.3** (FTPL Regret). *For any sequence $\{D_t\}_{t=1}^{T}$ selected by the learner, for $T \geq \max\{\frac{32C^2|S|^6}{\gamma^2}, \frac{2C^2}{\gamma^2} \ln(\frac{2|\mathcal{G}|}{\delta})\}$ with probability at least $1 - \delta$,*

$$\sum_{t=1}^{H} U(D_t, \lambda_t) - \min_{\lambda \in \Lambda} \sum_{t=1}^{H} U(D_t, \lambda) \leq \gamma T \ .$$

*Proof.* Proof provided in Appendix A.2.

□

**Corollary 3.4** (Tabular Apx-Minimax). *With probability at least $1 - \delta$, using Algorithm 1 with FTPL (Algorithm 2) converges to a $\nu$-approximate equilibrium in $poly(\frac{1}{\gamma}, \frac{1}{\delta}, |S|, \log|\mathcal{G}|)$.*

**Algorithm 2** FTPL in Tabular MDPs

1: **Input:** $\mathcal{G}$ groups, $r$ reward function, access to MDP $\mathcal{M}$, $\eta = \sqrt{\frac{C}{2|S|T}}$
2: Initialize cumulative losses $c_0(s) = 0 \;\forall s$
3: **for** $t = 1, \ldots, T$ **do**
4:     Receive $D_t$ from the Learner
5:     Sample a noise vector $N^t \sim \mathcal{U}([0, \frac{1}{\eta}]^{|S|})$
6:     Define $c^{<t}(s) = \sum_{i=0}^{t-1} c_i(s)$   *cumulative cost seen so far*
7:     $g_t \leftarrow$ Lin-OPT$(c^{<t} + N^t)$
8:     Assign $\lambda_{g,t}$ based on the selection of $g_t$ (i.e. $Ce_k$ for min group $k$ from $g_t$, 0 elsewhere)
9:     Sample $(s_t, a_t)$ by executing a trajectory under $D_t$ and returning $(s_h, a_h)$, for $h \sim \mathcal{U}([H])$
10:    Observe $c_t(s_t) = r(s_t, a_t)$
11: **end for**

---

*Proof.* For $\gamma \leq \frac{\nu}{2}$ and $\epsilon \leq \frac{\nu}{2}$, this satisfies the no-regret condition necessary for Theorem 3.2 to hold in the corresponding time specified by Lemma 3.3     □

### 3.2. Large State Space MDPs and Groups with Separator Sets

When we are no longer in the tabular setting (and therefore cannot directly run FTPL over $\mathcal{S}$), it is computationally inefficient to run FTPL over a very large collection of groups. Therefore, we utilize a contextual variant of FTPL (Syrgkanis et al., 2016; Dudík et al., 2020), which is computationally efficient in very large action spaces. We briefly recall the setting of contextual FTPL, and formulate our learning problem in this setting. At each time step $t$, the adversary (learner) selects a context $\sigma_t \in \Sigma$ and and action $y_t \in \mathcal{Y}$. The regulator then chooses a policy $\psi_t \in \Psi$ and receives payoff $f(\psi(\sigma_t), y_t)$. To capture our reinforcement learning problem, we take $\Sigma = \mathcal{S}$, $\mathcal{Y} = [0, 1]$, and $\Psi = \mathcal{G}$. The choices of the adversary at round $t$ will be determined by the policy $D_t$ learned by the best response oracle at round $t$. The adversary selects its context and action by first sampling $h \sim \mathcal{U}([H])$ and executing a trajectory $\tau$ under $D_t$ to obtain $(s_h, a_h)$. The adversary then selects its action to be $y_t = r(s_t, a_t)$, for $a_t = D_t(s)$. The payoff function received by the regulator is then $f(g(s_t), r(s_t, a_t)) = g(s_t)r(s_t, a_t)$.

The contextual FTPL algorithms of (Syrgkanis et al., 2016; Dudík et al., 2020) are efficient even in our setting of large state spaces and large $\mathcal{G}$ only under the assumption of separator sets for the class of groups $\mathcal{G}$.

**Definition 3.5** (Separator (Goldman et al., 1993; Syrgkanis et al., 2016)). *A set $X \subseteq \mathcal{S}$ is a separator for groups $\mathcal{G}$ if for any two groups $g, g' \in \mathcal{G}$ there exists a context $s \in \mathcal{S}$ such that $g(s) \neq g'(s)$. For a separator $X$ of size $|X| = d$, we use $x_{1:d}$ to denote a sequence of separator set elements.*

**Algorithm 3** Contextual FTPL in Large State Space MDPs

1: **Input:** $\mathcal{G}$ groups, $x_{1:d}$ sequence of separator elements, $r$ reward function, access to MDP $\mathcal{M}$, $\rho = \sqrt{\frac{\log|\mathcal{G}|}{Td^{1/2}}}$
2: Initialize $\lambda_0 = 0$
3: **for** $t = 1, \ldots, T$ **do**
4:     Receive $D_t$ from the Learner
5:     Draw $\eta_j \sim$ Lap$(\rho)$ independently for $j = 1, ..., d$
6:     Let $s_{t-1}^+ = s_{1:t-1} \| x_{1:d}$ be the sequence of contexts played thus far, concatenated with the sequence of separator set elements
7:     Let $y_{t-1}^+ = y_{1:t-1} \| \eta_{1:d}$ be the sequence of actions played thus far, concatenated with the sequence of noise elements
8:     $g_t \leftarrow$ OPT$(s_{t-1}^+, y_{t-1}^+)$
9:     Assign $\lambda_t$ based on the selection of $g_t$
10:    Sample $(s_t, a_t)$ by executing a trajectory under $D_t$ and returning $(s_h, a_h)$ for $h \sim \mathcal{U}([H])$.
11:    Observe $y_t = r(s_t, a_t)$ and receive $y_t g_t(s_t)$
12: **end for**

---

Neel et al. (2019) note that many discrete concept classes that are well-studied in the PAC learning literature in $d$-dimensions like boolean conjunctions, disjunctions, parities, and halfspaces defined over the boolean hypercube have separator sets of size $d$. In our setting, we make use of the fact that we can represent group membership in terms of conjunctions over the $d$-dimensional feature space.

**Lemma 3.6** (Contextual FTPL Regret). *For any sequence of costs $c_t$, for $T \geq O(\frac{C^2 d^{3/2} \log|\mathcal{G}|}{\gamma^2} + \frac{C^2}{\gamma^2} \ln(\frac{|\mathcal{G}|}{\delta}))$, with probability at least $1 - \delta$, the expected regret of FTPL is*

$$\sum_{t=1}^{H} U(D_t, \lambda_t) - \min_{\lambda \in \Lambda} \sum_{t=1}^{H} U(D_t, \lambda) \leq \gamma T \;.$$

*Proof.* Proof provided in Appendix A.3.     □

**Corollary 3.7** (Contextual FTPL Apx-Minimax). *With probability at least $1 - \delta$, using Algorithm 1 with Contextual FTPL (Algorithm 3) converges to a $\nu$-approximate equilibrium in $poly(\frac{1}{\gamma}, \frac{1}{\delta}, d, \log|\mathcal{G}|)$.*

*Proof.* For $\gamma \leq \frac{\nu}{2}$, $\epsilon \leq \frac{\nu}{2}$, Theorem 3.2's no-regret conditions are satisfied in the times specified by Lemma 3.6.   □

### 3.3. Large State Space MDPs with General Groups

For large state spaces with general group structure, we utilize Fair and Fictitious Play (Follow the Leader (FTL) vs FTL) (Brown, 1949). Fictitious Play is easier to implement as it is a deterministic algorithm and it only requires one oracle call per player. However, it is only theoretically known to converge to an equilibrium in the limit.

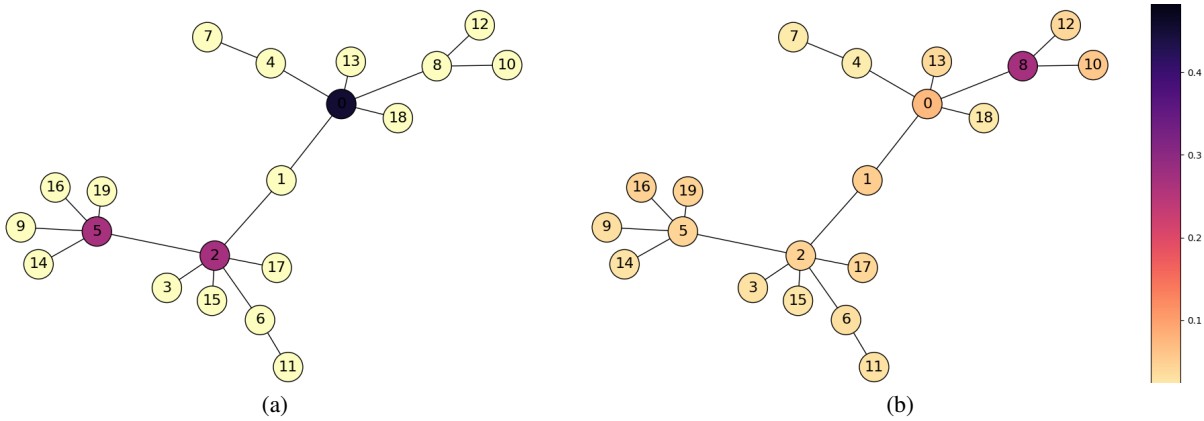

*Figure 1.* Depiction of the Barabási-Albert graph we use as an MDP. Nodes correspond to states and actions are deterministic moves to other nodes. Every state also has a self-loop action to remain in place. The groups are assigned depending on the number of outgoing edges. All nodes with 1 or 2 outgoing edges are in Group 0, nodes with 3 outgoing edges are in Group 1 and all others are in Group 2. Rewards $r(s, a)$ are assigned as 0.1 for all $s$ in Group 1, 0.2 for all nodes in Group 1 and 0.3 for nodes in Group 2. The start state is a random node in the graph in every episode. Figure (a) shows the occupancy distribution of the (non-fair) optimal policy. As we can see, the non-fair policy's goal is to quickly get to one of the nodes with 5 edges (Group 2) and stay there indefinitely to accumulate reward. After running our FairFictRL algorithm in Figure (b), the distribution over nodes is almost evenly spread across all nodes. The only outlier is node 8, as it is the only node that belongs to Group 1 and thus requires a large visitation number to satisfy our constraints.

---

**Algorithm 4** FairFictRL

1: **Input:** bound $C$, best-response error $\epsilon$, $\mathcal{G}$ groups, $r$ reward function, access to MDP $\mathcal{M}$
2: Initialize $\lambda_0^g = 0$ and $\hat{\lambda}_0^g = \lambda_0^g$
3: Initialize $\hat{D}_t$ by selecting a policy in $\Pi$
4: **for** $t = 1, \ldots, T$ **do**
5:    $D_t \in \text{FTL}(\hat{\lambda}_{t-1}, C)$ using $\mathcal{O}_\epsilon(\cdot, \hat{\lambda}_{t-1})$    returns $\max_D U(D, \hat{\lambda}_{t-1})$
6:    $\lambda_t = \text{FTL}(\hat{D}_{t-1}, C)$ using $Best_\lambda(\hat{D}_{t-1})$    returns $\max_\lambda U(\hat{D}_{t-1}, \lambda)$
7:    $\hat{D}_t = \frac{1}{t}\sum_{t'=0}^{t} D_{t'}$; $\hat{\lambda}_t = \frac{1}{t}\sum_{t'=0}^{t} \lambda_{t'}$
8: **end for**
9: **return** $(\hat{D}_T, \hat{\lambda}_T)$

---

**Lemma 3.8** (FairFict Convergence). *(Robinson, 1951) Fair-FictPlay converges in the limit to an equilibrium.*

Robinson (1951) showed that fictious play converges in the limit to equilibrium. Daskalakis & Pan (2014) showed that the convergence can be exponentially slow in the worst-case, at least with adversarially selected tie-breaking rules. Nevertheless, in non-adversarial settings it appears to be a practical algorithm. We use this algorithm in our experiments and also empirically demonstrate that it performs well, despite other algorithms' stronger theoretical guarantees.

## 4. Experimental Results

The experimental section focuses on an evaluation of the FairFictRL algorithm. FairFictRL is a very easy to implement version of our framework and even though it only has asymptotic guarantees, we will demonstrate that it can quickly converge to desirable solutions in practice.

### 4.1. MDP construction

Preferential attachment graphs are a commonly used model of social networks exhibiting a "rich get richer" distribution of connectivity. We construct a multi-objective RL task based on such graphs using the Barabási-Albert model (Barabási & Pósfai, 2016). in which nodes with higher degree tend to have a higher probability of forming connections with newer nodes on the network. In rounds, one new node is added to the graph until the final number of nodes is reached. Each new node forms $n_e$ new connections with probability proportional to the degree of existing nodes. Using this model, we construct the following graph MDP. The state space corresponds to the nodes on the graph and actions to edge selection on the current node. There is a scalar reward of $r(s, a)$ each time an edge is selected from a node. Each node is assigned to possibly several intersecting groups based on its degree. A learner is trying to traverse the graph with the goal of maximizing total cumulative reward, while maintaining that all groups receive at least average reward $\alpha/H$. The graph we consider is depicted in Figure 1.

## 4.2. FairFictRL Evaluation

First, we run FairFictRL with a minimum value of average group reward that we want to achieve of $\frac{\alpha}{H} = 0.04$. Given the reward stucture in the graph, it is not the case that all groups will obtain this reward under the optimal policy. We use standard value iteration as an oracle for the learner; for higher-dimensional problems, this could be substituted with a deep RL algorithm. The regulator runs the policy returned by the learner for 500 episodes to obtain an average reward estimate and best responds according to Definition 2.3 with $Ce_k = 25$. The results are shown in Figure 2.

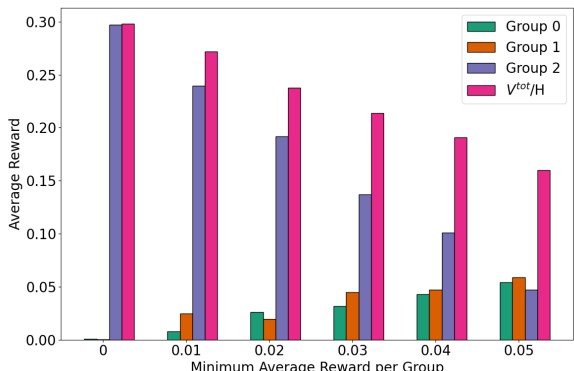

*Figure 3.* Total average reward compared to per-group average reward for varying $\alpha$. As the fairness constraint increases, total reward decreases while group rewards equal out until eventually all groups obtain the same average reward.

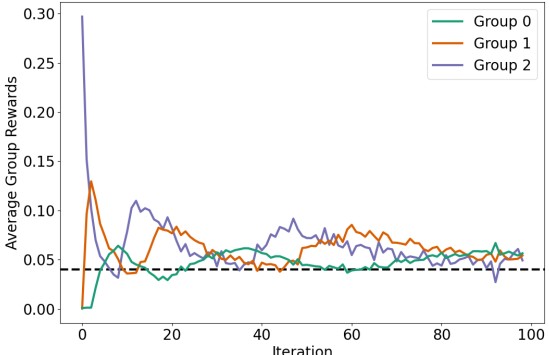

*Figure 2.* Average reward to groups during a run of FairFictRL on the MDP depicted in Figure 1 for $\frac{\alpha}{H} = 0.04$. The optimal (non-fair) behavior in the MDP is to move to any Group 2 node and stay there indefinitely, which achieves at most $0.3$ average reward. As we run FairFictRL, the learned mixture policy quickly ensures that all groups obtain at least $\frac{\alpha}{H}$ average reward.

## 5. Conclusion and Future Work

We consider MDPs with possibly an exponentially large number of intersecting groups over states. For solving a class of corresponding constrained or multi-objective optimization problems over these very large number of objectives, we provide several oracle-efficient algorithms. Namely, we provide algorithms for tabular MDPs and large state-spaces with separator sets over groups. In these approaches, we reduce the learner's problem to that of standard RL with one scalarized objective to learn a near-optimal policy. We reduce the regulator's problem to one of linear optimization. We also provide a version of Fictitious Play for large state-spaces with general group functions that has provable asymptotic guarantees. In practice, our experiments showed that they converge quickly for the preferential attachment graphs provided in our paper.

**Limitations and Future Work** Of course, each of these reductions relies on our ability to efficiently implement our Best Response and Optimization oracles. Moreover, many of our results are reliant on boolean group structure. It would be interesting to extend our results beyond exponentially large group classes to infinite group function classes. We also hope to apply these results to the setting of Reinforcement Learning with Human Feedback, especially with access to more fine-grained subgroup data.

As we can see, after only a few iterations, all groups obtain 0.04 average reward, which is consistently upheld throughout the execution of the algorithm. Furthermore, we demonstrate in Figure 1 that this corresponds to a more evenly distributed occupancy of the graph.

Finally, we analyze the Pareto frontier between fairness and maximum cumulative reward that can be obtained by varying values of $\alpha$ in Figure 3. The results highlight that by increasing the minimum average reward that we enforce, our algorithm equalizes the rewards between all groups more an more. This comes at the expense of total reward $V^{tot}$ which highlights the trade-off between fairness and optimality.

## Acknowledgements

This research was partially supported by the Simons collaboration on algorithmic fairness, NSF grants FAI-2147212 and CCF-2217062, the Army Research Office under MURI award W911NF201-0080, the DARPA Triage Challenge under award HR00112420305, and by the University of Pennsylvania ASSET center. Any opinions, findings, and

conclusion or recommendations expressed in this material are those of the authors and do not necessarily reflect the view of DARPA, the Army, or the US government. We would also like to thank Alejandro Ribeiro for his discussions with us. We would also like to thank Michael Dinitz for helpful conversations with us.

## Impact Statement

This paper provides largely theoretical work on algorithms to give guarantees of RL algorithms to a large number of possibly intersecting groups in both the tabular and large state-space setting. We hope that this work will be used to consider more seriously the guarantees to more marginalized groups of standard sequential-decision making algorithms and hope that more work is built to consider a variety of guarantees that ensure fairness in the form or definition that is best for each application. We do not feel that there are any major societal consequences (especially of a negative nature) that need to be highlighted here.

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

# A. Appendix

### A.1. Proof of Theorem 3.2

We want to show that $\frac{1}{T}\sum\lambda_t, \frac{1}{T}\sum D_t$ converge to an approximate minimax solution of the game if we have that

$$\sum_{t=1}^{T}\mathbb{E}\left[U(\hat{D}_t,\hat{\lambda}_t)\right] - \min_{D\sim 1_{\Pi}}\sum_{t=1}^{T}\mathbb{E}\left[U(D,\hat{\lambda}_t)\right] \leq BR_L(T,C,\epsilon)$$

and that

$$\max_{\lambda\in\Lambda}\sum_{t=1}^{T}\mathbb{E}\left[U(\hat{D}_t,\lambda)\right] - \sum_{t=1}^{T}\mathbb{E}\left[U(\hat{D}_t,\hat{\lambda}_t)\right] \leq reg_A(T,C,\gamma)$$

which are sublinear in $T$. We want to show that:
First for any $\lambda$,

$$\text{want to show: } U(\hat{D}_T,\lambda) \geq U(\hat{D}_T,\hat{\lambda}_T) - \frac{1}{T}reg_A(T,C,\gamma) - \epsilon \tag{5}$$

$$U(\hat{D}_T,\lambda) = U\left(\frac{1}{T}\sum_{t=1}^{T}D_t,\lambda\right) \quad \text{(linear in } D\text{)} \tag{6}$$

$$= \frac{1}{T}\sum_{t=1}^{T}U(D_t,\lambda) \tag{7}$$

$$\geq \frac{1}{T}\sum_{t=1}^{T}(U(D_t,\lambda_t)) - \frac{1}{T}reg_A(T,C,\gamma) \quad \text{(because no-regret)} \tag{8}$$

$$\geq \frac{1}{T}\sum_{t=1}^{T}U(\hat{D}_t,\lambda_t) - \frac{1}{T}reg_A(T,C,\gamma) - \epsilon \quad \text{(because apx-best response)} \tag{9}$$

$$= U(\hat{D}_T,\hat{\lambda}_T) - \frac{1}{T}reg_A(T,C,\gamma) - \epsilon \quad \text{(because linearity)} \tag{10}$$

$$\tag{11}$$

Second for any $D$,

$$\text{want to show: } U(D,\hat{\lambda}_T) \leq U(\hat{D}_T,\hat{\lambda}_T) + \frac{1}{T}reg_A(T,C,\gamma) + \epsilon \tag{12}$$

$$U(D,\hat{\lambda}_T) = \frac{1}{T}\sum_{t=1}^{T}U(D,\lambda_t) \tag{13}$$

$$\leq \frac{1}{T}\sum_{t=1}^{T}U(D_t,\lambda_t) + \epsilon \quad \text{(because } D_t \text{ is apx-best response)} \tag{14}$$

$$\leq \frac{1}{T}\sum_{t=1}^{T}U(D_t,\hat{\lambda}_T) + \frac{1}{T}reg_A(T,C,\gamma) + \epsilon \quad \text{(because no-regret)} \tag{15}$$

$$= U(\hat{D}_T,\hat{\lambda}_T) + \frac{1}{T}reg_A(T,C,\gamma) + \epsilon \quad \text{(because linearity)} \tag{16}$$

Thus, for any $\lambda, D$:

$$U(D,\hat{\lambda}_T) - \frac{1}{T}reg_A(T,C,\gamma) - \epsilon \leq U(\hat{D}_T,\hat{\lambda}_T) \leq U(\hat{D}_T,\lambda) + \frac{1}{T}reg_A(T,C,\gamma) + \epsilon$$

So if $\nu \geq \frac{1}{T}reg_A(T,C,\gamma) + \epsilon$, then we have gotten an approximate minimax solution using (Freund & Schapire, 1996). Note that this algorithm makes $O(T)$ best-response oracle queries and $O(T)$ Lin-OPT queries for the learner and regulator respectively.

## A.2. Proof of Lemma 3.3

*Proof.* (Kalai & Vempala, 2005) and (Roth, 2024), FTPL's regret can be bounded by

$$\left| \frac{1}{T} \sum_{t=1}^{T} (r(s_t, a_t)) g_t(s_t) - \min_{g \in \mathcal{G}} \frac{1}{T} \sum_{t=1}^{T} (r(s_t, a_t)) g(s_t) \right| \leq \frac{\sqrt{8|S|^3}}{\sqrt{T}}$$

For For $T \geq \frac{8C^2|S|^6}{\gamma_1^2}$, we have that

$$\left| \frac{1}{T} \sum_{t=1}^{T} (r(s_t, a_t)) g_t(s_t) - \min_{g \in \mathcal{G}} \frac{1}{T} \sum_{t=1}^{T} (r(s_t, a_t)) g(s_t) \right| \leq \frac{\gamma_1}{C}$$

Recall the definition of $V^g(\pi_t) = H \, \mathbb{E}_{t \in [H]} \, \mathbb{E}_{P,\pi,\mu}[(r(s_t, a_t) g(s_t)]$. Define $Y_t = \frac{1}{T}((r(s_t, a_t)) g_t(s_t) - (r(s_t, a_t)) - \frac{1}{TH}((V^{g_t}(\pi_t)) - (V^g(\pi_t)))$. We can show that the difference sequence $\{Y_t\}_{t=1}^{T}$ forms a Martingale Difference Sequence. Therefore $\sum_t Y_t$ also forms a Martingale. Using Azuma-Hoeffding's inequality, for any $\lambda \in \Lambda$

$$Pr\left( \left| \frac{1}{T}(r(s_t, a_t)) g_t(s_t) - (r(s_t, a_t))) g(s_t)) - \frac{1}{TH}((V^{g_t}(\pi_t)) - (V^g(\pi_t))) \right| \geq \frac{\gamma_2}{C} \right) \leq 2 \exp \frac{-2T\gamma_2^2}{C^2}$$

Union bounding over all groups, we want

$$2|\mathcal{G}| \exp \frac{-2T\gamma_2^2}{C^2} \leq \delta.$$

Therefore, for $T \geq \frac{C^2}{2\epsilon_2^2} \ln(\frac{2|\mathcal{G}|}{\delta})$, with probability at least $1 - \delta$, for all groups $g \in \mathcal{G}$,

$$\left| \frac{1}{T}((r(s_t, a_t)) g_t(s_t) - \min_{g \in \mathcal{G}}(r(s_t, a_t)) g(s_t)) - \frac{1}{TH}((V^{g_t}(\pi_t)) - \min_{g \in \mathcal{G}}(V^g(\pi_t))) \right| \leq \frac{\gamma_2}{C}$$

Therefore,

$$\left| \frac{1}{TH} \sum_{t=1}^{T} (V^{g_t}(\pi_t)) - \min_{g \in \mathcal{G}} \frac{1}{TH} \sum_{t=1}^{T} (V^g(\pi_t)) \right| \leq \frac{\gamma_1 + \gamma_2}{C}$$

Notice that

$$\frac{1}{T} \sum_{t=1}^{T} (r(s_t, a_t) g_t(s_t) -) - \min_{g \in \mathcal{G}} \frac{1}{T} \sum_{t=1}^{T} (r(s_t, a_t) g_t(s_t))$$

$$= (\frac{1}{T} \sum_{t=1}^{T} r(s_t, a_t) + r(s_t, a_t) g_t(s_t) - \frac{\alpha}{H}) - (\min_{g \in \mathcal{G}} \frac{1}{T} \sum_{t=1}^{T} r(s_t, a_t) + r(s_t, a_t) g(s_t) - \frac{\alpha}{H})$$

Similarly,

$$\left| \frac{1}{TH} \sum_{t=1}^{T} (V^{g_t}(\pi_t)) - \min_{g \in \mathcal{G}} \frac{1}{TH} \sum_{t=1}^{T} (V^g(\pi_t)) \right|$$

$$= (\frac{1}{TH} \sum_{t=1}^{T} \frac{1}{H} V^{tot}(\pi_t) + V^{g_t}(\pi_t) - \alpha) - (\min_{g \in \mathcal{G}} \frac{1}{TH} \sum_{t=1}^{T} \frac{1}{H} V^{tot}(\pi_t) + V^g(\pi_t) - \alpha)$$

$$\leq \frac{\gamma_1 + \gamma_2}{C}$$

To see the final connection with the objective $U$, we need to introduce $\lambda$. Observe that we could have defined the action space of FTPL in terms of $\lambda \in \Lambda$, where for every $g$, $\lambda^g \in \mathbb{R}^{|S|}$ and $\lambda^g(s) = Cg(s)$. That is, every selection of $g \in \mathcal{G}$ has a corresponding $\lambda^g \in \Lambda$ which puts weight $C$ on the dimension corresponding to the $gth$ group. It is important to distinguish different representations of $\lambda$. $\lambda \in \Lambda$ is a $|\mathcal{G}|$-dimensional vector over groups. $\lambda^g$, however, is an $|S|$-dimensional vector

over the state space for the selected group $g$. We acknowledge that we overload this notation because $\lambda^g$ without mention of a state can refer to the weight placed by the regulator on that group. Hence, for a given selected $\lambda \in \Lambda$, we use the notation $\lambda^g(s)$ to denote the corresponding group selected out of that regulator weight assignment and then its corresponding values over the state space. This notation makes explicit the implicit relationship between group selection and $\lambda$ selection. We can scale the above terms $r(s_t, a_t)g_t(s_t) - \frac{\alpha}{H}$ and $r(s_t, a_t)g(s_t) - \frac{\alpha}{H}$ by the corresponding $\lambda^{g_t}(s_t)$ or $\lambda^g(s_t)$ value To get $r(s_t, a_t)\lambda^{g_t}(s_t) - \frac{C\alpha}{H}$ and $r(s_t, a_t)\lambda^g(s_t) - \frac{C\alpha}{H}$ (notice that the vector $\lambda \in \mathbb{R}^{|\mathcal{G}|}$ is $C$ times a one-hot encoding). Similarly,

$$(\frac{1}{H}V^{tot}(\pi_t) + \sum_{g \in \mathcal{G}} \lambda^{g_t}(\frac{1}{TH}\sum_{t=1}^{T}(V^{g_t}(\pi_t) - \alpha))) - (\frac{1}{H}V^{tot}(\pi_t) + \min_{g \in \mathcal{G}}\sum_{g \in \mathcal{G}} \lambda^g(\frac{1}{TH}\sum_{t=1}^{T}(V^g(\pi_t) - \alpha)))$$

$$\leq \gamma_1 + \gamma_2$$

Finally,

$$\frac{1}{T}\sum_{t=1}^{H}U(D_t, \lambda_t) - \min_{\lambda \in \Lambda}\frac{1}{T}\sum_{t=1}^{H}U(D_t, \lambda) \leq \gamma$$

for $\gamma_1, \gamma_2 = \frac{\gamma}{2}$ and noting that $\pi_t$ is the policy specified by $D_t$. $\qquad\square$

### A.3. Proof of Lemma 3.6

*Proof.* We utilize (Syrgkanis et al., 2016) and (Dudík et al., 2020) for our regret bound and construction. (Syrgkanis et al., 2016) prove that for Laplace noise $\mathrm{Lap}(\rho)$ and groups $\mathcal{G}$ with separator $X$ of size $d = |X|$, their contextual FTPL algorithm enjoys regret

$$\left|\frac{1}{T}\sum_{t=1}^{T}(r(s_t, a_t))g_t(s_t) - \min_{g \in \mathcal{G}}\frac{1}{T}\sum_{t=1}^{T}(r(s_t, a_t))g(s_t)\right| \leq O\left(\frac{\rho d}{T}\sum_{t=1}^{T}\mathbb{E}\left[\max_{g \in \mathcal{G}}(g(s_t)r(s_t, a_t))^2\right] + \frac{\sqrt{d}\log(|\mathcal{G}|)}{\rho T}\right)$$

$$\leq O\left(\frac{d^{3/4}\sqrt{\log|G|}}{\sqrt{T}}\right)$$

for optimal choice of $\rho$, and runs in time $\mathrm{poly}(d, T, \log(|\mathcal{G}|))$. Therefore, for the below condition to hold, we would need $T \geq \frac{C^2 d^{3/2}\log|\mathcal{G}|}{\gamma_1^2}$.

We will use similar notation and arguments to the proof of Lemma 3.3. For $T \geq \frac{C^2 d^{3/2}\log|\mathcal{G}|}{\gamma_1^2}$,

$$\left|\frac{1}{T}\sum_{t=1}^{T}(r(s_t, a_t))g_t(s_t) - \min_{g \in \mathcal{G}}\frac{1}{T}\sum_{t=1}^{T}(r(s_t, a_t))g(s_t)\right| \leq \frac{\gamma_1}{C}$$

Once again, we use a similar argument to the proof of Lemma 3.3, Using Azuma-Hoeffding's inequality for a given $\lambda$,

$$Pr\left(\left|\frac{1}{T}(r(s_t, a_t))g_t(s_t) - (r(s_t, a_t))g(s_t)) - \frac{1}{TH}((V^{g_t}(\pi_t)) - (V^g(\pi_t)))\right| \geq \frac{\gamma_2}{C}\right) \leq 2\exp\frac{-2T\gamma_2^2}{C^2}$$

Union bounding over all groups, we want

$$2|\mathcal{G}|\exp\frac{-2T\gamma_2^2}{C^2} \leq \delta.$$

Therefore, for $T \geq \frac{C^2}{2\gamma_2^2}\ln(\frac{2|\mathcal{G}|}{\delta})$, with probability at least $1 - \delta$, for all groups $g \in \mathcal{G}$,

$$\left|\frac{1}{T}((r(s_t, a_t))g_t(s_t) - \min_{g \in \mathcal{G}}(r(s_t, a_t))g(s_t)) - \frac{1}{TH}((V^{g_t}(\pi_t)) - \min_{g \in \mathcal{G}}(V^g(\pi_t)))\right| \leq \frac{\gamma_2}{C}$$

Therefore,

$$\left|\frac{1}{TH}\sum_{t=1}^{T}(V^{g_t}(\pi_t)) - \min_{g \in \mathcal{G}}\frac{1}{TH}\sum_{t=1}^{T}(V^g(\pi_t))\right| \leq \frac{\gamma_1 + \gamma_2}{C}$$

Finally using a very similar argument to the proof of Lemma 3.3 to relate to $\lambda$,

$$\frac{1}{T}\sum_{t=1}^{H}U(D_t,\lambda_t) - \min_{\lambda\in\Lambda}\frac{1}{T}\sum_{t=1}^{H}U(D_t,\lambda) \le \gamma$$

for $\gamma_1,\gamma_2 = \frac{\gamma}{2}$. □

## A.4. Error Cancellations

Recall the constrained RL problem:

$$\max_{D\in\Delta\Pi}\quad \mathbb{E}_{\pi\sim D}[V^{tot}(\pi)]$$
$$\text{subject to}\quad \mathbb{E}_{\pi\sim D}[V^g(\pi)] \ge \alpha,\quad \forall g\in\mathcal{G}$$

In this section, we will use the shorthand $AV^{tot}(\pi)$ to denote the average reward rather than $V^{tot}(\pi)$ and $AV^g(\pi)$ for the average reward to group $g$. Now, we will also define $AV^{tot}(D) = \mathbb{E}_{\pi\sim D}[AV^{tot}(\pi)]$ and $AV^g(D) = \mathbb{E}_{\pi\sim D}[AV^g(\pi)]$.

Now define $h^g(D) = \frac{\alpha}{H} - AV^g(D)$ and $f(D) = AV^{tot}(D)$. Now, we can rewrite the constrained RL problem as:

$$\max_{D\in\Delta\Pi}\quad f(D)$$
$$\text{subject to}\quad h^g(D) \le 0,\quad \forall g\in\mathcal{G}$$

Now, notice that $h^g(D)$ is affine (and therefore convex) in $D$ and 0 is a constant so $\max\{h^g(D),0\}$ is convex in $D$. Therefore, we can rewrite our optimization problem as:

$$\max_{D\in\Delta\Pi}\quad f(D)$$
$$\text{subject to}\quad \max\{h^g(D),0\} \le 0,\quad \forall g\in\mathcal{G}$$

Notice that the feasibility sets over $D$ are the same between these two problems. Now, we can write the following Lagrangian.

$$\mathcal{L}(D,\lambda) = f(D) - \sum_{g\in\mathcal{G}}\lambda^g(\max\{h^g(D),0\}).$$

The conditions of Sion's minimax theorem still hold for this Lagrangian. Observe that for this objective, when considering the notion of average violations, one does not face the issue of error cancellations because the max term is always nonnegative. For the learner, we can define a new scalarized reward function in terms of the regulator's selection of $\lambda$ for which it can utilize the standard approximate best response oracle. Observe now however, that $k^g(D)$ is now convex in $D$ rather than linear, but we still need to provide estimates of $\max\{h^g(D),0\}$.

Define $k(x) = \max\{x,0\}$. We want to bound $|k(h^g(D)) - k(\hat{h}^g(D))|$. Notice, that to get our good estimate of $\hat{h}^g(D_t))$, we will sample from this procedure and average the following: sample a policy from $D_t$ and generate a trajectory, then uniformly sample a time step $h$ and receive the corresponding $s_h, a_h, r(s_h, a_h)$. Now, we will average a sufficiently large number of terms $\frac{\alpha}{H} - r(s_h,a_h)g(s_h)$ to get an estimate for $\hat{h}^g(D)$ within $\epsilon'$ using Hoeffding's inequality. Union-bounding over groups, we want $2|\mathcal{G}|e^{-2n\epsilon'^2} \le \delta'$, which requires $n \ge \frac{1}{\epsilon'^2}\frac{\ln(2|\mathcal{G}|)}{\delta'}$. We would like $\epsilon' \in O(\frac{1}{C\sqrt{T}})$ (because of its contribution to overall regret). Therefore, we would like $n \ge T\frac{\ln(2|\mathcal{G}|)}{\delta'}$. Now assume that with probability at least $1-\delta'$, for all $g\in\mathcal{G}$, $(\hat{h}^g(D) - h^g(D)) \le \epsilon'$. It is important to note however, that since the objective is now concave in $D$, rather than linear.

$$r_\lambda(s_t,a_t) = \left(r(s_t,a_t) - \sum_{g\in\mathcal{G}}\lambda^g\max\{\hat{h}^g(D),0\}\right)$$

Notice that this reward has some error from the "true" scalarized reward and therefore, our learner will be approximate best responding.

**Theorem A.1** (Approximate Min-Max Error Cancellation). *For a sequence of distributions over best-response policies $(D_1, ..., D_T)$ and Lagrangian weights from no-regret updates $(\lambda_1, ..., \lambda_T)$ maintained by the learner and regulator, MORL-BRNR [Algorithm 1] returns a $\nu$-approximate saddle point solution $(\hat{D}, \hat{\lambda})$ of $\mathcal{L}$.*

*Proof.* First for any $\lambda$,

$$\text{want to show: } \mathcal{L}(\hat{D}_T, \lambda) \geq \mathcal{L}(\hat{D}_T, \hat{\lambda}_T) - \frac{1}{T} reg_A(T, C, \gamma) - \epsilon \tag{17}$$

$$\mathcal{L}(\hat{D}_T, \lambda) = \mathcal{L}\left(\frac{1}{T}\sum_{t=1}^{T} D_t, \lambda\right) \quad \text{(linear in } D\text{)} \tag{18}$$

$$\geq \frac{1}{T}\sum_{t=1}^{T} \mathcal{L}(D_t, \lambda) \quad \text{(concavity in} D\text{)} \tag{19}$$

$$\geq \frac{1}{T}\sum_{t=1}^{T} (\mathcal{L}(D_t, \lambda_t)) - \frac{1}{T} reg_A(T, C, \gamma) \quad \text{(because no-regret)} \tag{20}$$

$$\geq \frac{1}{T}\sum_{t=1}^{T} \mathcal{L}(\hat{D}_t, \lambda_t) - \frac{1}{T} reg_A(T, C, \gamma) - \epsilon \quad \text{(because apx-best response)} \tag{21}$$

$$= \mathcal{L}(\hat{D}_T, \hat{\lambda}_T) - \frac{1}{T} reg_A(T, C, \gamma) - \epsilon \quad \text{(because linearity)} \tag{22}$$

$$\tag{23}$$

Second for any $D$,

$$\text{want to show: } \mathcal{L}(D, \hat{\lambda}_T) \leq \mathcal{L}(\hat{D}_T, \hat{\lambda}_T) + \frac{1}{T} reg_A(T, C, \gamma) + \epsilon \tag{24}$$

$$\mathcal{L}(D, \hat{\lambda}_T) = \frac{1}{T}\sum_{t=1}^{T} \mathcal{L}(D, \lambda_t) \tag{25}$$

$$\leq \frac{1}{T}\sum_{t=1}^{T} \mathcal{L}(D_t, \lambda_t) + \epsilon \quad \text{(because } D_t \text{ is apx-best response)} \tag{26}$$

$$\leq \frac{1}{T}\sum_{t=1}^{T} \mathcal{L}(D_t, \hat{\lambda}_T) + \frac{1}{T} reg_A(T, C, \gamma) + \epsilon \quad \text{(because no-regret)} \tag{27}$$

$$\leq \mathcal{L}(\hat{D}_T, \hat{\lambda}_T) + \frac{1}{T} reg_A(T, C, \gamma) + \epsilon \quad \text{(because concavity)} \tag{28}$$

Thus, for any $\lambda, D$:

$$\mathcal{L}(D, \hat{\lambda}_T) - \frac{1}{T} reg_A(T, C, \gamma) - \epsilon \leq \mathcal{L}(\hat{D}_T, \hat{\lambda}_T) \leq \mathcal{L}(\hat{D}_T, \lambda) + \frac{1}{T} reg_A(T, C, \gamma) + \epsilon$$

So if $\nu \geq \frac{1}{T} reg_A(T, C, \gamma) + \epsilon$, then we have gotten an approximate minimax solution using (Freund & Schapire, 1996). Note that this algorithm makes $O(T)$ best-response oracle queries and $O(T)$ Lin-OPT queries for the learner and regulator respectively.

Furthermore, notice that if we wanted to return one policy rather than $\bar{D}$ (deterministic), we could evaluate our selection of policies and based upon the best performing one could make a probabilistic argument for how likely that single policy is to simultaneously satisfy all constraints. $\square$

### A.4.1. GENERALIZED FTPL

We still get our regret bound with respect to $\gamma$ for a sufficient polynomial sample complexity.

---

**Algorithm 5** Contextual FTPL Error Cancellation

---

1: **Input:** $\mathcal{G}$ group functions, $x_{1:d}$ sequence of separator elements, $r$ reward function, access to MDP $\mathcal{M}$, $\rho = \sqrt{\frac{\log|\mathcal{G}|}{Td^{1/2}}}$ number of samples per iteration $n$
2: Initialize $\lambda_0 = 0$
3: **for** $t = 1, \ldots, T$ **do**
4:     Receive $D_t$ from the Learner
5:     Draw $\eta_j \sim \text{Lap}(\rho)$ independently for $j = 1, ..., d$
6:     For all $g \in \mathcal{G}$, let $g_{x_{1:d}} = (g_t(x_1), \ldots, g_t(x_d))$ in the following:
    Select $g_t$ such that $\sum_{i=0}^{t-1} c_i(g_t) + \eta \cdot g_{t,x_{1:d}} \leq \sum_{i=0}^{t-1} c_i(g) + \eta \cdot g_{x_{1:d}} + \epsilon$ for all $g$
7:     Assign $\lambda_t$ based on the selection of $g_t$
8:     Generate $n$ samples of $\{s_i, a_i, r(s_i, a_i)\}_{i=1}^n$ by creating $n$ independent trajectories of $D_t$ and uniformly sampling from $[H]$ in each one and computing an estimate of average reward and define $y_{t,i} = r(s_i, a_i)$
9:     Receive $c_t(g_t) = \max\{\frac{\alpha}{H} - \frac{1}{n}\sum_{i=1}^n y_{t,i}g(s_i), 0\}$
10: **end for**

---

**Lemma A.2** (Contextual FTPL Regret). *For any sequence of costs $c_t$, for $T \geq O\left(\frac{C^2 d^{3/2}\log|\mathcal{G}|}{\gamma_1^2} + \frac{C^2}{\gamma_2^2}\ln(\frac{|\mathcal{G}|}{\delta})\right)$, with probability at least $1 - \delta$, the expected regret of FTPL is*

$$\sum_{t=1}^H \mathcal{L}(D_t, \lambda_t) - \min_{\lambda \in \Lambda} \sum_{t=1}^H \mathcal{L}(D_t, \lambda) \leq \gamma T$$

*Proof.* Although the sample complexity of our result changes–(now we require $nT$ samples rather than purely $T$ earlier), the regret bound from Lemma 3.6 should still hold with $\gamma_1 = \gamma_2 = \frac{\gamma - \epsilon'}{2}$. Also observe $\delta' = \frac{\delta}{2}$. $\square$

**Corollary A.3** (Contextual FTPL Apx-Minimax with Error Cancellations). *With probability at least $1 - \delta$, using Algorithm 1 with Generalized FTPL (Algorithm 5) converges to a $\nu$-approximate equilibrium in $poly(\frac{1}{\gamma}, \frac{1}{\delta}, d, \log|\mathcal{G}|)$.*

*Proof.* For $\gamma \leq \frac{\nu}{2} - \epsilon'$ and $\epsilon \leq \frac{\nu}{2} + \epsilon'$, this satisfies the no-regret condition necessary for Theorem A.1 to hold in the corresponding time specified by Lemma A.2 $\square$

A.4.2. FICTIOUS PLAY

Notice that for Ficitious Play, all the convergence results discussed for Algorithm 4 still hold.

---

**Algorithm 6** FairFictRL Error Cancellation

---

1: **Input:** bound $C$, best-response error $\epsilon$, $\mathcal{G}$ groups, $r$ reward function, access to MDP $\mathcal{M}$
2: Initialize $\lambda_0^g = 0$ and $\hat{\lambda}_0^g = \lambda_0^g$
3: Initialize $\hat{D}_t$ by selecting a policy in $\Pi$
4: **for** $t = 1, \ldots, T$ **do**
5:     $D_t \in \text{FTL}(\hat{\lambda}_{t-1}, C)$ using $\mathcal{O}_\epsilon(\cdot, \hat{\lambda}_{t-1})$    returns $\max_D \mathcal{L}(D, \hat{\lambda}_{t-1})$
6:     $\lambda_t = \text{FTL}(\hat{D_{t-1}}, C)$ using $Best_\lambda(\hat{D}_{t-1})$    returns $\max_\lambda \mathcal{L}(\hat{D}_{t-1}, \lambda)$
7:     $\hat{D}_t = \frac{1}{t}\sum_{t'=0}^t D_{t'}$; $\hat{\lambda}_t = \frac{1}{t}\sum_{t'=0}^t \lambda_{t'}$
8: **end for**
9: **return** $(\hat{D}_T, \hat{\lambda}_T)$

---

