# OpenReview forum: "Intersectional Fairness in Reinforcement Learning with Large State and Constraint Spaces"
_ICML.cc/2025/Conference — ICML 2025 poster_

### Official Review · Reviewer_Gqfm · 2025-03-11

**Overall Recommendation:** 3

**Summary:**

The paper addresses intersectional fairness in reinforcement learning (RL) with exponentially many constraints arising from overlapping groups. The authors propose oracle-efficient algorithms for three settings: 1) tabular MDPs, 2) large MDPs with structured groups (via separator sets), and 3) general groups (using Fictitious Play). The key idea is to reduce the constrained RL problem to a minimax game between a learner (optimizing policies) and a regulator (enforcing group fairness), leveraging optimization oracles to handle large constraint sets. Theoretical guarantees include sublinear regret and convergence to approximate equilibria. Experiments on a preferential attachment graph MDP demonstrate that their method ensures minimum reward guarantees for all groups, balancing fairness and total reward.

**Claims And Evidence:**

see the comments below

**Essential References Not Discussed:**

NA

**Experimental Designs Or Analyses:**

see the comments below

**Methods And Evaluation Criteria:**

see the comments below

**Other Comments Or Suggestions:**

NA

**Other Strengths And Weaknesses:**

Strengths:

- Novel integration of optimization oracles for exponentially many constraints.

- Theoretical contributions in oracle-efficient RL with intersectional fairness.

Weaknesses:

- Practical reliance on strong oracles (e.g., Lin-OPT) without discussing implementability.

- Limited empirical scope (one synthetic MDP).

- No discussion of computational complexity for large-scale deployments.

**Questions For Authors:**

- The theory supports exponential groups, but experiments use only three. How does the performance scale with $d$? A response showing linear/polynomial scaling would strengthen the claim.
- How does the method compare to the baseline methods?

**Relation To Broader Scientific Literature:**

see the comments below

**Theoretical Claims:**

see the comments below

---

> ### Author Rebuttal · Authors · 2025-03-31
>
> Dear Reviewer Gqfm,
>
> Thank you for your positive assessment of our work.
>
> **Empirical Scope**
> As we highlight in Section 5 on Future Work, we completely agree that moving towards more realistic scenarios with these algorithms poses an interesting future direction for our work. However, the current manuscript is largely a theoretical contribution and we believe that moving to more complex oracle settings will come with its own challenges and open questions. These include studying the effects of potentially sub-optimal oracles in practice, which will likely warrant a different type of publication. Our experiments primarily aim to verify the effectiveness of our FairFictRL algorithm. This algorithm is easy to implement but it does not come with with polynomial-time guarantees. Thus, our experiments validate that this easy version of the algorithm might be a feasible alternative in practice.
>
> **Optimization Oracle Assumption** Our optimization oracle assumption is  mainly stating that we are able to do linear optimization efficiently in practice. Algorithms to achieve this exist in practice. For example, in the case of continuous space, interior point methods or barrier methods work well and in the case of conjunctions, integer linear programs and mixed integer programs, etc. work effectively in practice. For best response oracles one may hope to resort to deep reinforcement learning algorithms that may return policies close to optimal. We are happy to extend the discussion on this by adding text akin to the previous sentences to the paper.
>
> **Computational Complexity and Scaling** Since we provide oracle efficient algorithms, the run time (which we bound in our lemmas) is the core component of our computational complexity. Our algorithms mainly make O(T) oracle calls total, comprising the majority of the computation within each round; the only other computation is the sampling of noise vectors of size $O(d)$ or $O(|S|)$ at each round for the Follow the Perturbed Leader base algorithms. We will update our paper to include a statement highlighting the computational complexity of our algorithms. Lemma $3.3$ shows that the computational complexity of our algorithm for tabular MDPs scales logarithmically in $|\mathcal{G}|$ and polynomially in $|S|$. Lemma $3.6$ shows that the computational complexity of our algorithm for continuous spaces and groups with separator structor scales with the size of the separator set $d$ and $\ln(|\mathcal{G}|)$. We believe that the impact of the number of groups on scaling to larger settings in practice would be negligible.
>
> **Baseline Methods**
> We appreciate the question. We have not implemented any baselines in this setting and we are not aware of any baselines that would be suitable. We view the main claim of the experimental section as validating that our proposed FairFictRL algorithm works. We do not believe that the addition of baselines would strengthen this claim substantially.

---

### Official Review · Reviewer_zwEM · 2025-03-13

**Overall Recommendation:** 3

**Summary:**

This paper investigates an interesting problem of intersectional fairness in RL. Unlike standard fairness approaches in RL, this work formulates fairness as a multi-objective optimization problem. Specifically, the goal is to optimize fairness and efficiency by maximizing the utility of the least advantaged group when demographic groups are available. The authors propose oracle-efficient algorithms for solving both tabular and large-scale MDPs. Theoretical results are provided, and experiments on a synthetic graph MDP validate the approach.

**Claims And Evidence:**

Most of the paper claims are well-supported through theoretical results and empirical validation in synthetic graph MDP. However, the claim that the method achieves Pareto dominance in the synthetic graph MDP is questionable. Optimizing only for the minimum group utility ignores the needs of other groups, which may sacrifice overall efficiency and Pareto optimality.

**Essential References Not Discussed:**

The paper adequately discusses related work.

**Experimental Designs Or Analyses:**

The paper lacks fair RL baseline comparisons. While related works (e.g., Satija et al., 2023) are cited, they are not included as baselines in the experiments. The experimental evaluations are also limited. I believe testing the method on real-world environments such as college admissions, healthcare, or loan approval etc would significantly strengthen the paper.

**Methods And Evaluation Criteria:**

While the paper addresses an important gap in fair RL, its practical applicability and scalability require further investigations. The experiments are limited to simple, small, tabular MDP, which may not generalize well to more complex RL settings.

**Other Comments Or Suggestions:**

See above.

**Other Strengths And Weaknesses:**

Strengths:

* The paper introduces an important and novel problem of intersectional fairness in RL.
* The theoretical results, particularly the no-regret guarantees, are well-establish and provide enough justification for the proposed method.
* The paper proposed oracle-efficient algorithms that scale to both tabular and large-scale MDPs.
* The approach adapts fictitious play from classification fairness to RL, demonstrating its applicability to sequential decision-making with large state spaces.

Weaknesses:
* The paper focuses on group fairness (specifically minimax) but does not consider individual fairness, which is often more practical in RL and real-world applications. Could the approach be extended to an individual fairness setting?
* The minimax objective optimizes for the worst-off group but may sacrifice total utility and fail to ensure proportional or Pareto-optimal fairness.
* The experimental evaluations are only in a synthetic MDP with only three groups, which does not reflect the complexity of real-world RL applications with a large number of demographic groups.
* Moreover, the paper also lacks comparisons to fair RL methods.

**Questions For Authors:**

* Can the proposed method accommodate other fairness notions, such as leximin or proportional fairness, beyond minimax?
* How does the proposed method scale with a higher number of objectives (e.g., 10)? In such cases, the number of intersectional groups could be 1024. Would the approach remain computationally tractable?
* How would Lin-OPT/OPT be approximated in practice for large group sizes?
* Are there relevant baseline methods that could be included for comparison, both in terms of fairness and efficiency?
* How does the method generalize to real-world RL environments, where state/action spaces are large or continuous?

**Relation To Broader Scientific Literature:**

The paper makes a significant theoretical contribution by introducing intersectional fairness in RL, a relatively unexplored area. It adapts a technique from classification fairness (fictitious play) to sequential decision-making in RL and demonstrate its applicability to RL. While the theoretical insights are valuable, additional empirical validation would enhance its impact.

**Theoretical Claims:**

The theoretical results, particularly the no-regret guarantees in Theorems 3.1 and 3.2, appear correct to me. The discussion in the main paper is clear, but I have not verified the full proofs in the appendix.

---

> ### Author Rebuttal · Authors · 2025-03-31
>
> Dear Reviewer zwEM,
>
> Thank you for acknowledging the significant theoretical contribution of our work and the detailed feedback on our submission.
>
> **Individual Fairness** The notion of fairness we consider represents a middle ground between fully statistical notions of fairness and fully individual notions. By considering a sufficiently rich set of subgroups (rather than just the $d$ individual attributes or all individuals), we are able to provide algorithms that are tractable even for large state spaces while having meaningful guarantees for more fine-grained portions of the population. We believe this to be a valuable step towards methods that deal with the great variety of preferences and characteristics that individuals can exhibit, reflecting the vast diversity found within human sub-populations.
>
> **Pareto Optimal Fairness** This is a great point. You are correct in that the minimax optimization does not imply pareto optimality. However, what we can claim is that the minimax solution Pareto-dominates the solution that tries to equalize rewards across groups. We will clarify this in the text. We also do agree (as Figure 3 shows) that overall efficiency or optimality may be sacrificed when trying to optimize with fairness constraints in mind. This may unfortunately be unavoidable in many settings.
>
> **Empirical Scope**
> As we highlight in Section 5, we completely agree that more realistic scenarios pose an interesting future direction. However, the current manuscript is largely a theoretical contribution and we believe that moving to more complex oracle settings will come with its own challenges and open questions. These include studying the effects of potentially sub-optimal oracles, which will likely warrant a different type of publication. Our experiments primarily aim to verify the effectiveness of our FairFictRL algorithm. This algorithm is easy to implement but it does not come with polynomial-time guarantees. Thus, our experiments validate that this algorithm is a feasible alternative in practice.
>
> **Baseline methods**
> We appreciate the feedback on this. We view the main claim of the experimental section to be that our proposed FairFictRL algorithm is practical and we do not believe that the addition of baselines would strengthen this claim substantially.
>
> **Other Notions of Fairness**
> This is a great point. We agree that other notions like leximin fairness are very interesting. In fact, an idea for future work might be to adapt our method to iteratively solve for leximin fairness as in [1]. However, even in this classification setting, leximin fairness is a very subtle and technically unstable concept. In addition, it can become computationally challenging when the number of groups is large as one may have to iterate over all groups. Instead, our method is able to handle a broad class of linear/convex fairness constraints or objectives and can be implemented efficiently. That being said, there are still many other interesting fairness notions that one could explore and we are excited about doing so in the future.
>
> **Group/Objective Scaling** Our proposed method scales with the log of the number of groups as can be seen in Lemma $3.3$ for tabular MDPs and Lemma $3.6$ for large state spaces. This is one of the key strengths of our contribution as it allows us to handle exponentially many groups.
>
> **Approximate Lin-OPT**
> Our optimization oracle assumption is mainly stating that we are able to do linear optimization efficiently in practice.
> For this, interior point methods or barrier methods work well in the case of continuous spaces and in the case of conjunctions, integer linear programs and mixed integer programs work effectively. We are happy to extend the discussion on this by adding text akin to the previous sentences to the paper.
>
> **Large and continuous state-action spaces**
> Our methods already give provable guarantees for large spaces as we are able to make use of Algorithm $3$, which gives poly-time guarantees when we have group separator structure, or Algorithm $4$ for spaces without any group structure assumptions, but with an asymptotic guarantee. Here, one challenge would be to implement the policy oracle as mentioned before. One could hope to approximate solutions via standard deep RL techniques, which in many settings may be feasible. Yet, in various problems these algorithms may not return optimal policies in every execution [2]. As stated before we believe that moving to more complex oracle settings will come with its own challenges that lie beyond the scope of this manuscript.
>
> [1] Lexicographically Fair Learning: Algorithms and Generalization. Emily Diana et al. FORC 2021.
>
> [2] Deep Reinforcement Learning that Matters. Peter Henderson et al. AAAI 2018.

---

> > ### Comment · Reviewer_zwEM · 2025-04-03
> >
> > Thanks for the detailed response to my comments. While most of my concerns have been addressed, I still believe that including baselines and conducting additional experiments would strengthen the paper. Since my current score already indicate a weak accept, I will maintain it in light of the authors' response.

---

### Official Review · Reviewer_sNea · 2025-03-14

**Overall Recommendation:** 3

**Summary:**

This paper studies a constrained RL problem in the episodic setting that generalizes the problem of minimax group fairness, allowing for exponentially many group fairness constraints due to the intersection of grouping functions. The authors focus on developing oracle-efficient algorithms for solving this problem in large state and constraint spaces. Their key contribution lies in providing solutions for three scenarios: tabular MDPs, large state space MDPs with separator sets assumed groups, and large state space MDPs with general groups. They propose two FTPL-based algorithms for the first two settings, and a Fictitious Play-based algorithm for the third. Theoretically, they provide regret bound for the first two settings. The effectiveness of the third algorithm is demonstrated empirically through a graph-based simulation.

**Claims And Evidence:**

The claims presented in the paper are supported by clear and convincing evidence.

**Essential References Not Discussed:**

No.

**Experimental Designs Or Analyses:**

The numerical studies would benefit from further experimentation. Specifically, given the paper's focus on large state and constraint spaces, experiments should be added to examine such scenarios. Additionally, simulation studies should be conducted to verify the validity of the two FTPL algorithms.

**Methods And Evaluation Criteria:**

The proposed method provides a reasonable solution to the problem.

**Other Comments Or Suggestions:**

It would be beneficial for readers to have a clearer explanation of how the separator assumption impacts the computational efficiency of Algorithm 3.

**Other Strengths And Weaknesses:**

Strengths:
       1. The paper studies a interest problem in fair RL with a large number of intersecting groups.
       2. The authors provide oracle-efficient algorithms that can handle a large number of constraints efficiently, which is crucial in practical where the number of groups can be very large.
       3.The graph-based simulation study is novel and easy to understand.

Weakness:
       1. The paper would benefit from additional numerical experimentation to further validate the proposed algorithms.

**Questions For Authors:**

1. Could you please clarify how Algorithm 4 is implemented in the numerical study? For instance, could you provide details on the implementation of the Best Response and Optimization oracles?

2.  If the MDP is not accessible, do you have any insights on how your algorithms could be adapted?

**Relation To Broader Scientific Literature:**

This work relates to prior research in fair RL and constrained RL. Rather than solving constrained RL problems by enumerating the constraints, the authors propose oracle-efficient algorithms.

**Theoretical Claims:**

I did not verify the correctness of the proofs.

---

> ### Author Rebuttal · Authors · 2025-03-31
>
> Dear Reviewer sNea,
>
> Thank you for your positive assessment of our work and your feedback.
>
> **Additional Experimental Simulations** The experimental study that we
> provide is used to highlight the applicability of our FairFictRL algorithm. Fictitious Play is easy to implement in practice because it only relies on Follow the
> Leader for both players and does not require noise generation that scales with the
> dimension of the action space. However it does not come with polynomial-time
> guarantees and our experiments aim to verify that it is still a good choice in
> practice. The Follow the Perturbed Leader-based algorithms themselves already
> have provable guarantees to converge, and we believe that adding additional
> experiments here may distract from the main message of the experimental section.
>
> **Optimization Oracles and Experiments** In our experiments we consider a
> tabular MDP, for which the best response oracle is implemented via value iteration. The
> regulator can select the group that minimizes the past cumulative cost by iterating over the groups,
> because the number of groups is small in this experiment; in practice would need to use a stronger
> optimization oracle that is able to do linear optimization efficiently. In the case
> of conjunctions, this might also involve Integer Linear Programs or Mixed Integer
> Programs. Regarding MDP access, none of our algorithms require direct access to the MDPs. We only assume access to the
> MDP in our experimental section, in which we use value iteration to implement our best response oracle.
> We will update the wording in the manuscript to make this more clear.
> Note that to implement the best response oracle, we simply need to be able to optimize a
> given specified reward function within an MDP, which can be done with standard
> RL methods. These do not need to depend on access
> to the MDP. We simply choose value iteration because the goal of the
> experimental section is not to evaluate different oracles but rather to evaluate
> our FairFictRL algorithm under the proposed assumptions.
>
> **Separator sets** The separator assumption directly affects our computational efficiency via the dependence of d in the bound given in Lemma 3.6.
> Since we are providing oracle-efficient algorithms, our primary computational-
> complexity analysis comes from the number of rounds our algorithm runs for
> given that we make O(T) oracle calls.

---

### Official Review · Reviewer_xri7 · 2025-03-16

**Overall Recommendation:** 3

**Summary:**

This paper explores the problem of intersectional fairness in reinforcement learning (RL) with large state and constraint spaces, proposing several oracle-efficient algorithms to optimize multiple objectives simultaneously while ensuring fairness across intersecting demographic groups. The authors provide theoretical guarantees for their algorithms and validate their effectiveness through experiments on a preferential attachment graph MDP.

**Claims And Evidence:**

good

**Essential References Not Discussed:**

no

**Experimental Designs Or Analyses:**

good

**Methods And Evaluation Criteria:**

good

**Other Comments Or Suggestions:**

see questions

**Other Strengths And Weaknesses:**

see questions

**Questions For Authors:**

This paper explores the problem of intersectional fairness in reinforcement learning (RL) with large state and constraint spaces, proposing several oracle-efficient algorithms to optimize multiple objectives simultaneously while ensuring fairness across intersecting demographic groups. The authors provide theoretical guarantees for their algorithms and validate their effectiveness through experiments on a preferential attachment graph MDP. I am unfamiliar with this specific area of research, so my comments may not be highly technical, but I have a couple of concerns.

Firstly, why did the authors not conduct experiments using deep reinforcement learning algorithms? Given that deep RL is widely used in complex, high-dimensional problems, it would be interesting to see how well the proposed methods perform in such settings. This could provide more insights into the scalability and practical applicability of the algorithms.

Secondly, I am interested in the potential applications of this work. The authors mention fairness in RL, which is a critical issue in many real-world scenarios. However, it would be helpful to discuss more specific scenarios where this algorithm could be particularly impactful.

**Relation To Broader Scientific Literature:**

good

**Theoretical Claims:**

good

---

> ### Author Rebuttal · Authors · 2025-03-31
>
> Dear Reviewer xri7,
>
> We appreciate the positive assessment of our work.
>
> **Deep RL experiments** As we highlight in  Section 5 on Future Work, we completely agree that deep learning poses an interesting future direction for this type of work. However, the current manuscript is largely a theoretical contribution and we believe that moving to deep learning will come with its own challenges and open questions such as studying the effects of potentially sub-optimal oracles in practice, which will likely warrant a different type of publication. Our experiments primarily aim to verify the effectiveness of our FairFictRL algorithm. This algorithm is very easy to implement but, unlike our other algorithms, it does not come with with polynomial-time guarantees. Yet, our experiments validate that this easy version of the algorithm might be a feasible alternative in practice.
>
>
> **Applications** Fairness in Reinforcement learning plays a large role in various applications such as healthcare or loan approval. However, we believe that one of the most interesting applications in our current timeline is the application to Reinforcement Learning from Human Feedback (RLHF). Our technique could be a step towards preventing LLMs from outputting discriminatory or toxic prompts to various (intersectional) groups. We will update the paper to include additional discussion of this application.

---

### Decision · Program_Chairs · 2025-05-01

**Decision:**

Accept (poster)

**Comment:**

This paper considers the problem of intersectional fairness in RL with large state and constraint spaces. All the reviewers agreed that the theoretical contribution is novel and interesting, particularly the design of oracle-efficient algorithms with a large number of objectives. The paper might benefit from improved experimental and/or simulation frameworks. However, I believe that the contribution of the paper is mainly theoretical.